# Engineering, decoding and systems-level characterization of chimpanzee cytomegalovirus

Quang Vinh Phan[1,¤], Boris Bogdanow[2], Emanuel Wyler[3], Markus Landthaler[3], Fan Liu[2], Christian Hagemeier[1], Lüder Wiebusch[1]*

1 Department of Pediatric Oncology/Hematology, Charité—Universitätsmedizin Berlin, Berlin, Germany, 2 Department of Structural Biology, Leibniz-Forschungsinstitut für Molekulare Pharmakologie, Berlin, Germany, 3 Berlin Institute for Medical Systems Biology, Max-Delbrück-Center for Molecular Medicine, Berlin, Germany

¤ Current address: Wyss Institute for Biologically Inspired Engineering, Harvard University, Boston, Massachusetts, United States of America

* lueder.wiebusch@charite.de

**Data Availability Statement:** The mass spectrometry proteomics data have been deposited in the ProteomeXchange Consortium with the data set identifier PXD027434. The RNA sequencing

## Abstract

The chimpanzee cytomegalovirus (CCMV) is the closest relative of human CMV (HCMV). Because of the high conservation between these two species and the ability of human cells to fully support CCMV replication, CCMV holds great potential as a model system for HCMV. To make the CCMV genome available for precise and rapid gene manipulation techniques, we captured the genomic DNA of CCMV strain Heberling as a bacterial artificial chromosome (BAC). Selected BAC clones were reconstituted to infectious viruses, growing to similar high titers as parental CCMV. DNA sequencing confirmed the integrity of our clones and led to the identification of two polymorphic loci and a deletion-prone region within the CCMV genome. To re-evaluate the CCMV coding potential, we analyzed the viral transcriptome and proteome and identified several novel ORFs, splice variants, and regulatory RNAs. We further characterized the dynamics of CCMV gene expression and found that viral proteins cluster into five distinct temporal classes. In addition, our datasets revealed that the host response to CCMV infection and the de-regulation of cellular pathways are in line with known hallmarks of HCMV infection. In a first functional experiment, we investigated a proposed frameshift mutation in UL128 that was suspected to restrict CCMV's cell tropism. In fact, repair of this frameshift re-established productive CCMV infection in endothelial and epithelial cells, expanding the options of CCMV as an infection model. Thus, BAC-cloned CCMV can serve as a powerful tool for systematic approaches in comparative functional genomics, exploiting the close phylogenetic relationship between CCMV and HCMV.

## Author summary

Human cytomegalovirus (HCMV) infection is associated with systemic disease in immunocompromised individuals and congenitally infected neonates. Animal CMVs and their

data have been deposited in the NCBI GEO database, with the accession number GSE171149. An annotated genome sequence of CCMV BAC-Phan9 is available at GenBank under the accession number MZ151943.

**Funding:** This work was supported by a grant (#900005) from the Joachim-Herz-Stiftung (https://www.joachim-herz-stiftung.de/en/) to LW. QVP was funded by a scholarship for postgraduate thesis projects (Charité Promotionsstipendium I) from the Charité Academics Grants Committee (https://nachwuchs.charite.de/en/). BB acknowledges funding from Deutsche Forschungsgemeinschaft (https://www.dfg.de/en/index.jsp) grant BO 5917/1-1. The funders had no role in study design, data collection and analysis, decision to publish, or preparation of the manuscript.

**Competing interests:** The authors have declared that no competing interests exist.

bacterial artificial chromosome (BAC) clones have been utilized as models for CMV infection and thereby contributed immensely to the understanding of pathogenesis, host immune response and underlying molecular mechanism of CMV infections. As the closest relative to HCMV, the chimpanzee CMV (CCMV) holds a great potential as a model system for HCMV infection but its application was limited due to the lack of tools and data for functional genomic analyses. Here, the cloning of the CCMV as a BAC vector made its viral genome available to gene targeting techniques that allow the efficient application of reverse genetic strategies. Furthermore, the multi-omic datasets created in this study provide an in-depth view of the viral gene repertoire and the host cell responses to infection, confirming the close phylogenetic relationship between HCMV and CCMV on a system level. Taken together, the newly established CCMV-BAC system presents a framework for HCMV modelling and comparative studies to address key questions in evolutionary processes and infection mechanisms.

## Introduction

Cytomegaloviruses (CMVs) are a group of β-herpesviruses infecting numerous primate species, including old and new world monkeys, great apes and humans [1]. As a result of prolonged periods of coevolution, CMVs are highly adapted to their specific hosts, making cross-species infections extremely rare events [2–4]. CMVs are very prevalent in their host populations, due to life-long, mostly asymptomatic infections. Medically most relevant is the human cytomegalovirus (HCMV) which causes serious complications in immunocompromised patients and neonates. Available HCMV treatment relies on inhibitors of viral DNA replication and packaging but their use is limited by side effects and the emergence of resistant virus strains. Currently, there are no approved HCMV vaccines available.

The severe consequences on health and the limitation in treatment options have motivated systematic efforts in understanding HCMV infection and replication mechanisms. Transcriptomic and proteomic studies have provided a comprehensive picture of the viral gene expression program [5–10]. This has been complemented by global assessment of virus-virus and virus-host protein interactions [11,12]. By genome-wide mutagenesis of viral DNA, a set of genes essential for HCMV replication *in vitro* has been identified, which mostly consists of "core" genes conserved throughout herpesviruses [13–15]. In addition, HCMV encodes an arsenal of accessory, less-conserved factors required for immune evasion [16,17], regulation of host metabolism [18], cell cycle control [19] and anti-apoptosis [20].

For the functional analysis of potential antiviral targets by reverse genetics, CMV genomes cloned as bacterial artificial chromosomes (BACs) serve as powerful tools [21,22]. First, BAC-captured CMV genomes are easily propagated and stably maintained as single clones in *Escherichia coli* (*E.coli*). Second, and more important, the large and complex CMV genomes become available for prokaryotic recombination techniques, allowing for efficient and reliable gene targeting. In addition to multiple HCMV strains cloned as BACs [23], animal CMV BACs are utilized as cloning platforms for *in vitro* and *in vivo* models and thereby provide the framework for a profound understanding of CMV infection and pathogenesis [24,25]. In particular, the rhesus CMV (RhCMV) BAC platform has become a popular model system for HCMV due to many shared features in infection, replication and immune control [26,27].

RhCMV proteins show only a moderate level of conservation towards HCMV homologs at the amino acid level [28]. Furthermore, in several genomic regions the genetic content of RhCMV diverges from HCMV due to gene duplications and losses that have occurred during

evolution of each species [25]. This degree of evolutionary divergence can set a limitation on the ability to assess HCMV gene functions by studying RhCMV homologs. By comparison, chimpanzee CMV (CCMV) shares a much higher level of similarity in both genome organization and protein content with HCMV [1,25,28,29]. The close phylogenetic relationship between HCMV and CCMV [30,31] is reflected by the fact that human cells are fully permissive for CCMV and vice versa [32,33]. Accordingly, whole genome sequencing data of the CCMV Heberling strain were utilized to re-evaluate the HCMV coding potential [33] and to investigate the adaptation of HCMV to its human host [1].

We envisioned CCMV to be an eminent model system that operates at the interface between animal and human CMV species and therefore BAC-cloned the CCMV Heberling genome. Deep sequencing of selected BAC clones and the genetically heterogeneous Heberling stock revealed hotspots for genetic variations within the CCMV genome. By analyzing the transcriptome and proteome of CCMV-BAC infected cells, we were able to expand the known coding content of CCMV, to delineate the dynamics of viral gene expression and the de-regulation of cellular pathways, revealing many parallels with known hallmarks of HCMV infection. Our system-level and functional characterization of a clonal CCMV strain thus reinforces the potential of CCMV as a model system for HCMV infection.

## Results

### BAC-cloning of CCMV Heberling strain

To make CCMV accessible for efficient genetic analyses, we set out to clone the CCMV strain Heberling genome as a BAC. Due to the high collinearity between HCMV and CCMV, we chose a cloning strategy similar to the approach taken for the generation of HCMV TB40-BAC4 [34]. To this end, the BAC donor plasmid pEB1097 [35] was modified in a way that integration of the BAC cassette via homologous recombination replaces the US2 to US6 region of CCMV (Fig 1A). After enrichment of recombined CCMV genomes by guanine phosphoribosyltransferase selection, circular viral DNA was extracted and transformed into *E. coli*. To check the overall integrity of the CCMV genome, randomly selected BAC clones were analyzed by AgeI restriction digest. Five clones closely matched the *in silico* predicted restriction pattern (Fig 1B) and were further analyzed for infectivity and virus growth. Upon re-transfection into primary human fibroblasts, all clones reconstituted to replication-competent viruses. Although their growth kinetics were similar to the parental Heberling virus, the BAC-derived viruses reached a lower maximum titer (Fig 1C).

Compared to the *in silico* prediction, two AgeI restriction fragments of 4,272 and 4,560 bp length were missing in the analytical digests of our CCMV BAC clones (see arrowheads in Fig 1B). As these fragments originate from the US region of the CCMV genome, we had to consider the possibility that the targeted integration of the BAC cassette was not as accurate as expected. We therefore performed polymerase chain reaction (PCR) assays and Sanger sequencing to validate the anticipated recombination event. In all clones we found an approximately 3.6 kbp genomic deletion adjacent to the 5'-end of the BAC cassette (S5A Fig) and an additional 5.2 kbp deletion in the vicinity of the 3'-end (S5B Fig). This finding was reminiscent of the accidental loss of the IRS1-US1 region that has often occurred during the generation of HCMV BAC clones [36,37].

To characterize the genomes of the selected CCMV BAC clones and the parental CCMV Heberling strain in more detail, we subjected them to whole genome sequencing. To our surprise, a substantial drop in sequencing read coverage was detected in the region between IRS1 and US10 of CCMV Heberling (Fig 2A). This indicates that a major proportion of genomes within the parental virus stock already carried a deletion in the IRS/US region prior to our

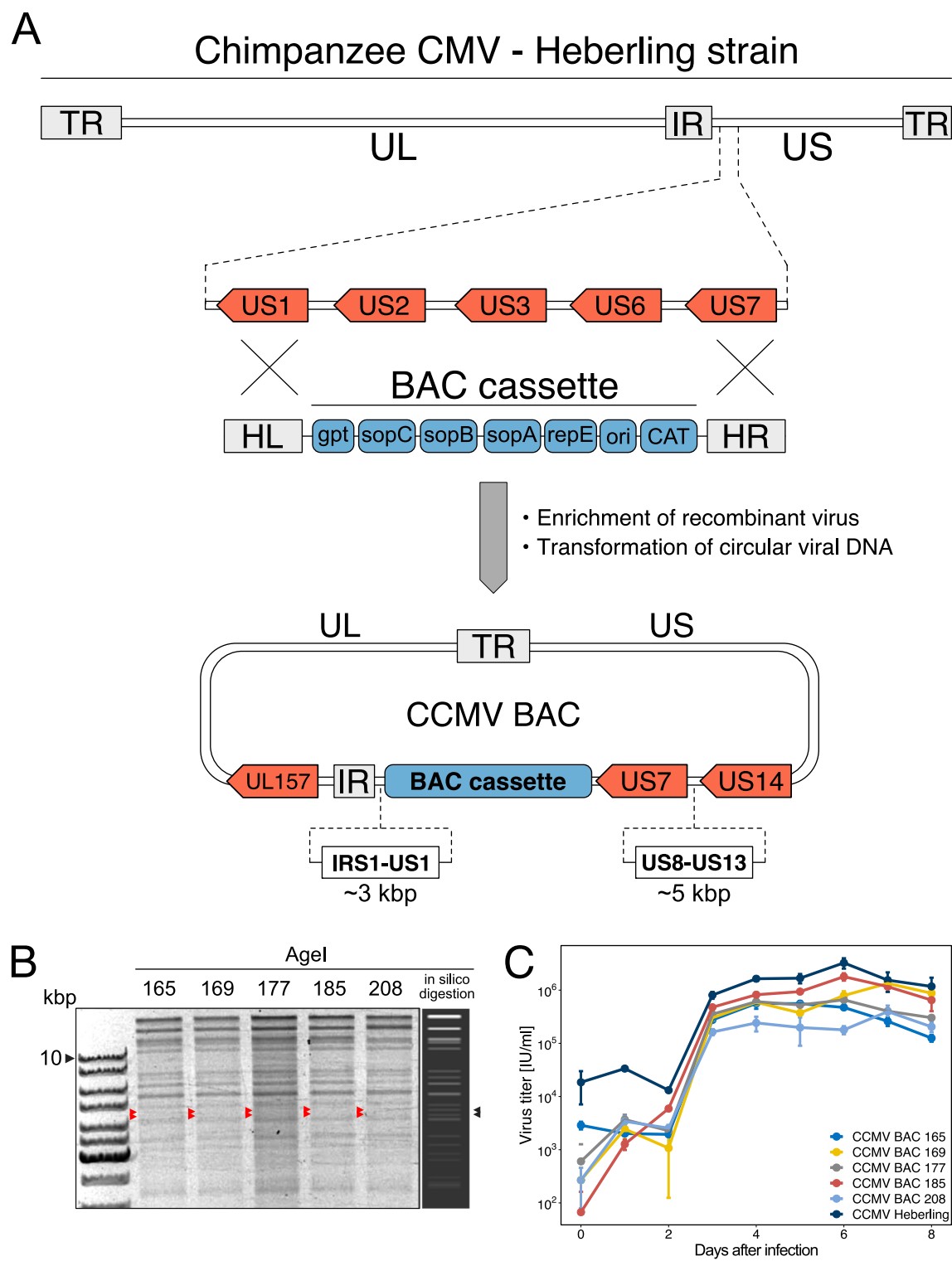

**Fig 1. BAC-cloning of CCMV strain Heberling.** (A) Targeted integration of the BAC cassette into the CCMV Heberling genome was mediated by homology arms left (HL) and right (HR), replacing the region between US1 and US7. The BAC cassette contains a replication origin (ori), a centromeric region (sopC), genes encoding replication (repE) and partitioning factors (sopA, sopB), as well as resistance genes for eukaryotic (xanthine phosphoribosyltransferase—gpt) and prokaryotic selection (chloramphenicol acetyltransferase—CAT). TR—terminal repeats; UL—unique long region; IR—internal repeats; US—unique short region. (B) Five BAC clones showed an AgeI restriction pattern matching the predicted *in silico* digest. Arrowheads indicate bands that were missing

due to an unintended deletion of IRS1 to US1 and US8 to US13 regions. (C) All selected CCMV BAC clones were infectious upon transfection into fibroblast but differ in their titer output performance. Means (center of the error bar) and standard errors of the mean of n = 3 are depicted.

cloning efforts. Importantly, targeting of this region was still possible as all clones showed integration of the BAC cassette in this particular genomic region (Fig 2B). Although alignment of the BAC sequencing reads to the reference genome confirmed the unintended deletion events of the IRS1-US1 and US8-US13 regions (S5A and S5B Fig), the overall CCMV genome integrity was maintained as no other major deletion or insertion (indels) events were detected (Figs 2B and S5C).

The whole genome sequencing data allowed an in-depth genotypic analysis of the CCMV BAC clones. In comparison to the published CCMV Heberling sequence [33], we observed an accumulation of single nucleotide variations (SNV) within the UL48-55 and the US26-28

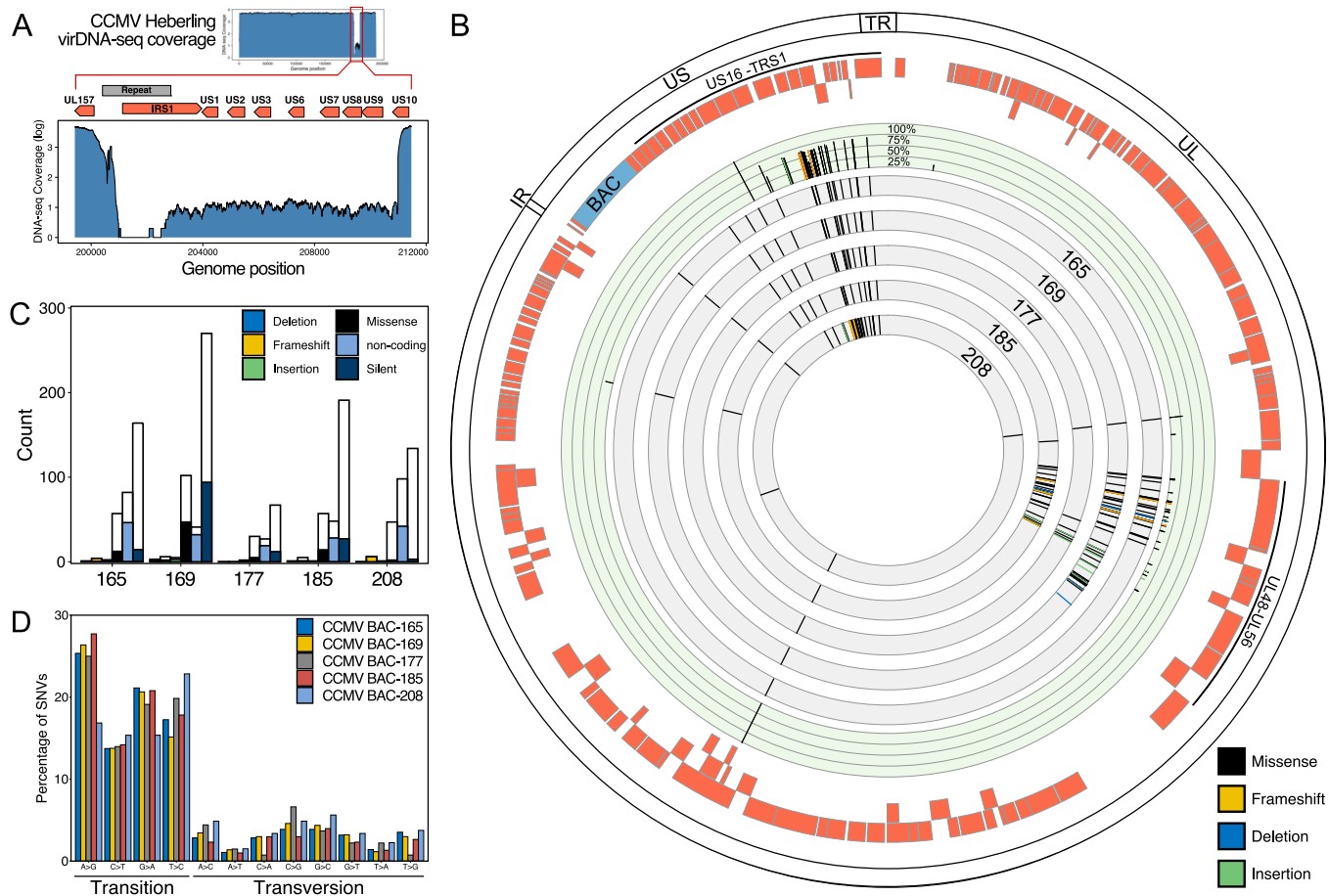

**Fig 2. Genotypic analysis of CCMV clones.** (A) Drop in sequencing read coverage indicates that a large proportion of viral genomes within the parental CCMV Heberling pool have a deletion event between IRS1 and US10. (B) Whole genome sequencing revealed variations in genomic sequences of selected CCMV BAC clones and the parental CCMV Heberling. Positions of missense, frameshift, deletion and insertion mutations for BAC clones are plotted in the grey tracks of the Circos plot. Intra-strain variations of the CCMV Heberling isolate are plotted in the green track; the frequency of each variation is indicated. Outer tracks: CCMV ORFs (red) and genomic regions. (C) The individual BAC clones were compared with the CCMV Heberling reference sequence for the absolute counts of single nucleotide variations (SNVs). White stacked bars indicate number of variations detected based on the alignment to the reference genome sequence of Heberling (NC_003521.1) but found in the re-sequenced parental CCMV virus stock pool. Colored bars indicate *de novo* variations in each clone that are neither detected in the original reference sequence nor the parental CCMV virus stock. CCMV BAC-177 showed the lowest number of SNVs and contained no frameshift mutations. (D) Similarly, the BAC clones were analyzed for the spectrum of transition and transversion mutations.

regions (Figs 2B and S6 and S1 Table). In contrast, CCMV Heberling, which had never been plaque purified [33], presented as a mixture of different genotypes. The nucleotide variations between these genotypes clustered at the same genomic positions as in our BAC clones (Fig 2B and S1 Table). In total, at least 62% (419/681) of SNV were carried over from the viral stock to our clones, leaving only a minor fraction of mutations that were likely to be created *de novo* during BAC generation. Silent mutations contributed the major proportion of SNVs in the analyzed CCMV sequences (Figs 2C and S6). This was in line with the observation that base transitions are more prevalent than base transversions (Fig 2D). The number of missense mutations ranged from 31 (BAC-177) to 102 (BAC-169) and 4 to 6 frameshift mutations were found in all BAC sequences except BAC-177. Taken together, BAC clone 177 was the most accurate representation of the originally published CCMV Heberling genome and we therefore chose this clone as a basis for further modifications.

## UL128 repair expands the cell tropism of CCMV

Fibroblasts are the preferred cell type for CMV isolation and propagation. However, extensive passage of HCMV in fibroblasts frequently results in disruptive mutations in genes of the pentameric complex (UL128, UL130, UL131A), rendering fibroblast-adapted virus strains unable to enter endothelial, epithelial and myeloid cells [23,38]. In consistence with this, a frameshift mutation within the first exon of the CCMV UL128 gene has been proposed for CCMV Heberling [33,39]. The proposed mutation was carried over to our BAC clones, but no further disruptive mutations were detected in the members of the pentameric complex. We suspected that the frameshift mutation restricts the cell tropism of CCMV and aimed to restore UL128 function by BAC mutagenesis. After removing the cytosine insertion at position 111 of the UL128 coding sequence (Fig 3A), we observed a substantial reduction in cell-free titer upon virus reconstitution in human fibroblasts. This was consistent with previous observations on HCMV-UL128 [40]. We then used low multiplicity of infection (MOI) conditions to infect human fibroblasts, epithelial and endothelial cells. In fibroblasts, the onset of viral gene expression occurred independent of the UL128 status. In contrast, only the UL128-repaired virus was capable of inducing IE gene expression and establishing virus growth in endothelial and epithelial cells (Fig 3B and 3C). This result confirmed the UL128 defect in the original CCMV Heberling isolate. By reinstating the endothelial cell tropism of CCMV, we expanded the options of our CCMV BAC for future experimental applications and provided a proof of concept that it is suitable for functional genomics approaches.

## Restoration of the US region

As mentioned above, all CCMV BAC clones displayed major deletions next to the BAC cassette integration site. Although the affected US1-US13 gene region is not essential for cytomegalovirus replication *in vitro* [41], it plays an important role in immune modulation [42]. To make CCMV BAC clone 177 useful for immunological research, we decided to restore the US region and make the BAC cassette excisable by Cre recombination (Fig 4A). The latter step was necessary to ensure that the BAC size does not exceed the packaging capacity of the viral capsid and to prevent the accumulation of unintended mutations around the prokaryotic vector sequences upon virus reconstitution [43].

While the seamless restoration of the US2-US13 region was readily accomplished by BAC mutagenesis, we did not succeed in re-introducing US1 and IRS1. Possibly, the close proximity to the internal repeat region resulted in promiscuous, non-specific recombination events. The BAC cassette was successfully flanked with loxP sites and upon virus reconstitution the Cre-mediated excision left only a single loxP site between US6 and US7 behind (Fig 4A). We verified the

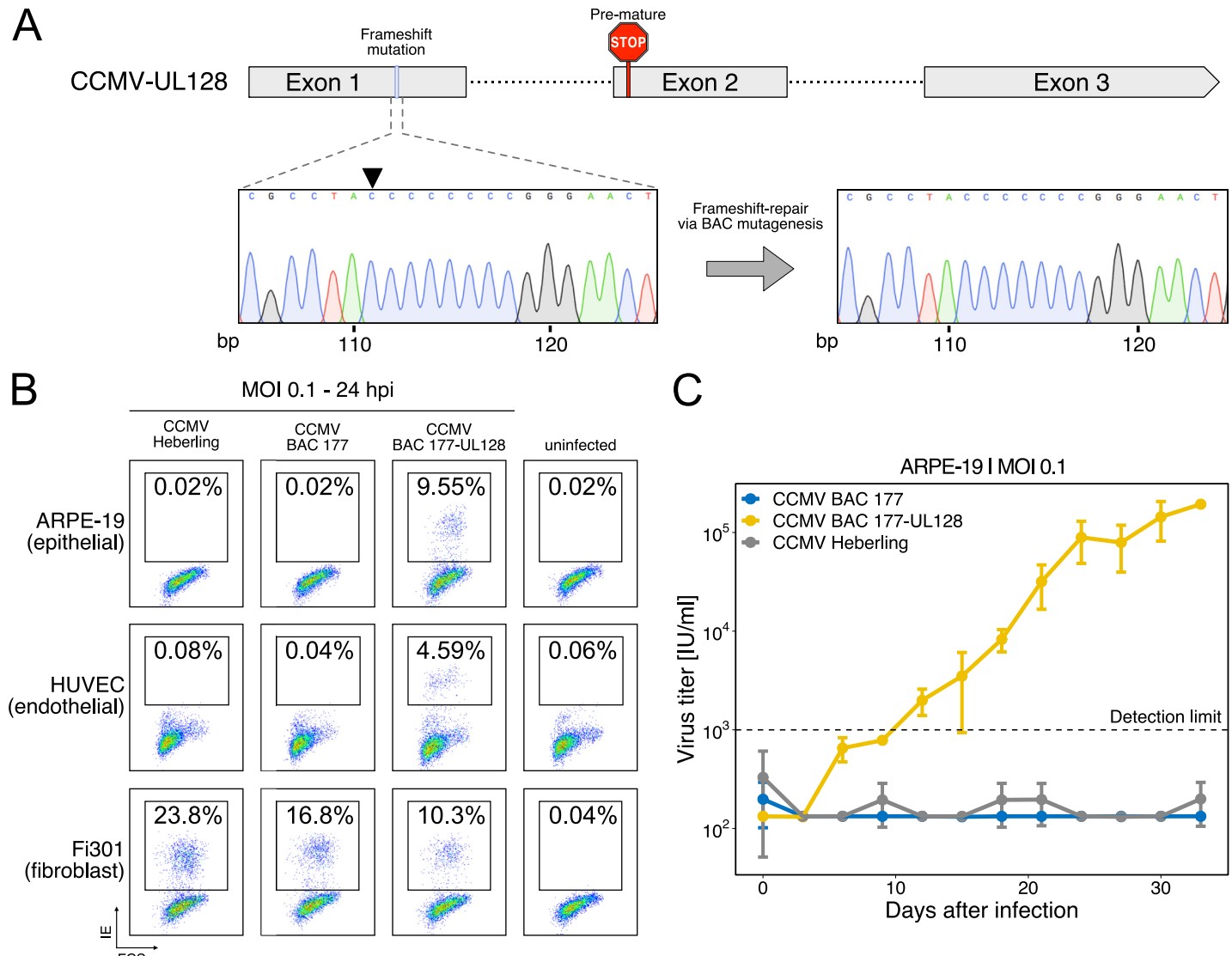

**Fig 3. UL128 repair rescues CCMV's tropism for epithelial and endothelial cells.** (A) In CCMV Heberling, a frameshift mutation in the first exon of UL128 leads to a premature stop codon in exon 2. Sanger sequencing chromatograms of BAC-177 and the UL128-repaired BAC-177 document the successful removal of the frameshift by BAC mutagenesis. (B, C) The indicated cell types were infected with viruses derived either from the Heberling isolate, BAC-177 or the UL128-repaired BAC-177, using an MOI of 0.1. At 24 h post infection, the cells were analyzed for IE1/2 gene expression by flow cytometry (B). Virus growth in epithelial cells was monitored over a 33-day period. Means (center of the error bar) and standard errors of the mean of n = 3 are depicted (C).

integrity of our improved CCMV BAC construct, named BAC-Phan9, via whole genome sequencing and detected only a few minor deviations from the adjusted CCMV BAC-177 reference sequence in the modified US region (Fig 4A). None of these changes had an impact on protein coding sequences and the BAC-Phan9 derived virus grew to similar titers as the parental CCMV BAC-177 (Fig 4B). Despite the missing US1/IRS1 sequences, CCMV BAC-Phan9 still displays all four genomic isoforms (S7 Fig). Furthermore, RNA sequence read coverage displays clear boundaries for US2, US3-US6, US7 and US8-US9 genes (S8 Fig), indicating that transcriptional termination of these genes is intact and functions IRS1/US1-independent.

In order to assess the ability of CCMV BAC-Phan9 to faithfully recapitulate the gene expression program of the parental CCMV Heberling strain, we performed a comparative

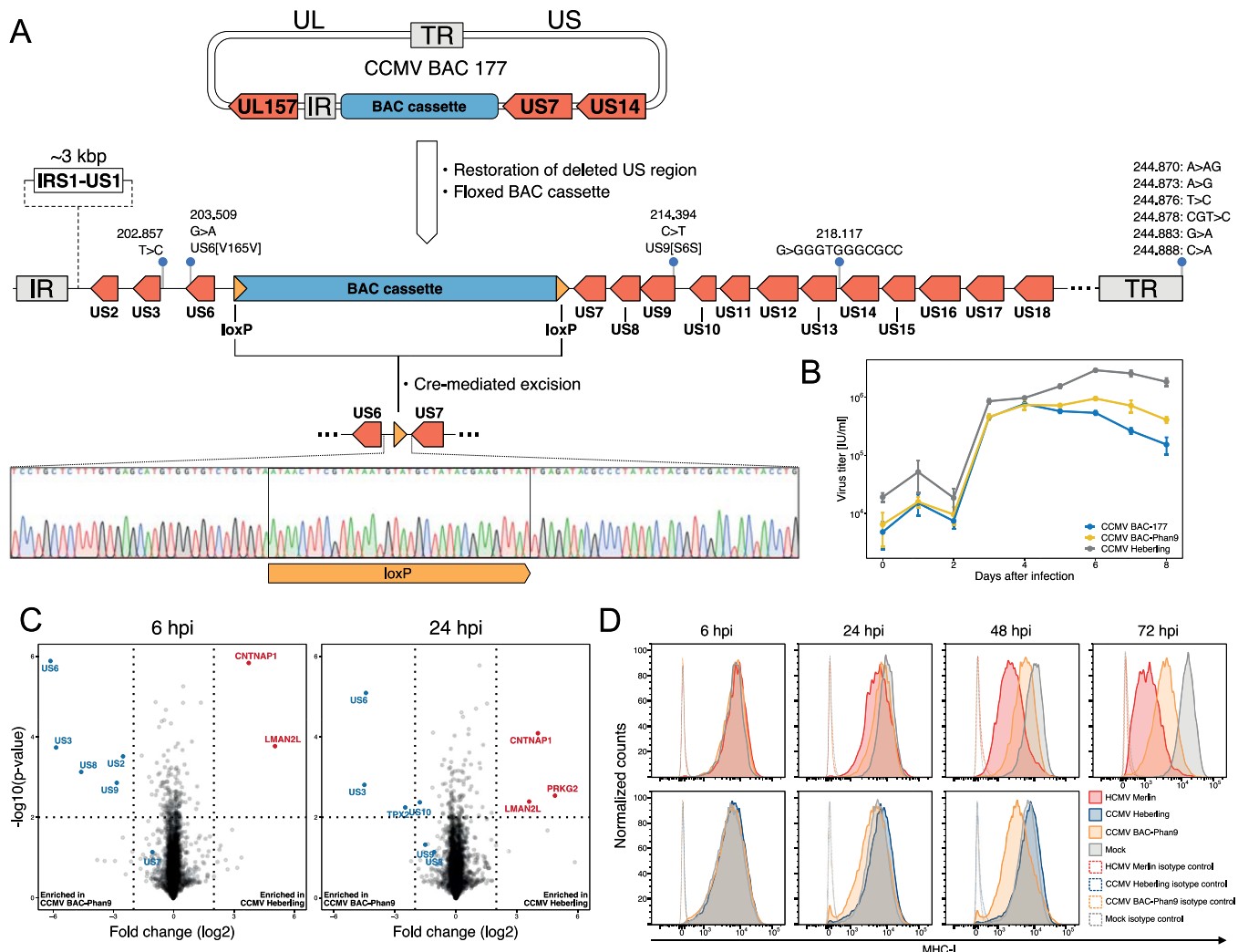

**Fig 4. Restoration of deleted US regions and floxing of BAC cassette.** (A) Schematic overview of US2-US6 and US8-US13 regions that were restored by BAC mutagenesis in CCMV BAC-177. Region IRS1 to US1 was not reinstated. LoxP sites were inserted at both ends of the BAC cassette to remove prokaryotic sequences by Cre recombinase expression during virus reconstitution. The final CCMV-BAC construct was named BAC-Phan9. Whole genome sequencing verified the integrity of BAC-Phan9. Coverage of DNA sequencing and positions of minor sequence variations in the US and TR regions are plotted. Sanger sequencing of DNA from reconstituted virus confirms the expected Cre-mediated recombination outcome at the US6-US7 region. (B) One-step growth curves of CCMV BAC-177, BAC-Phan9 and CCMV Heberling were analyzed in human fibroblasts. Means (center of the error bar) and standard errors of the mean of n = 3 are depicted. (C) Cells infected with either CCMV Heberling or BAC-Phan9 were analyzed for protein expression by mass spectrometry. The observed differences at 6 and 24 hpi are depicted as volcano plots. Proteins enriched in BAC-Phan9 infection and proteins orginitating from the US2-US10 region are labeled in blue, proteins enriched in Heberling infection in red. (D) Flow cytometry analysis of MHC-I surface expression during infection with CCMV BAC-Phan9, HCMV Merlin (upper row) and CCMV Heberling (lower row).

proteomic analysis of infected cells. Overall, the proteomic profiles of both infections were very similar (Fig 4C). However, the reinstated genes US2, US3, US6, US8 and US9 were significantly higher expressed in the BAC-Phan9 infection. This was consistent with our finding that a genomic IRS1-US10 deletion predominates in the Heberling virus population (Fig 2A). As US2, US3 and US6 gene products are known to prevent cell surface expression of the major histocompatibility complex class-I (MHC-I) [44], we also anticipated differences in this particular immune evasion capability and monitored MHC-I surface expression during CCMV infection. While CCMV Heberling had no effect, BAC-Phan9 was able to down-regulate MHC-I, although not to the same extent as HCMV Merlin (Fig 4D). This finding underlines

the importance of our cloning approach as the unstable CCMV genome could be fixed and repaired, making it available for functional studies. In the following, the BAC-Phan9 derived virus was utilized for a comprehensive characterization of the CCMV gene expression program.

## Evaluation of the CCMV coding capacity

Since the initial sequencing of the CCMV genome [33], the understanding of the genetic content of other CMV genomes has been dramatically improved [6,29]. In contrast, CCMV gene expression has never been analyzed systematically. In order to get a deeper understanding of CCMV infection and the evolutionary relationship of CCMV and HCMV, we took our cloned virus and analyzed the CCMV coding potential by state of the art transcriptomics and proteomics. The RNA and protein material we used for this purpose was isolated from productively infected cells at five different time points, covering the whole CCMV replication cycle. First, we created a preliminary annotation file of potential novel open reading frames (ORFs) by the following stepwise approach (Fig 5A): (i) The CCMV genome was scanned by tBLASTn [45] for homologs of previously identified non-canonical ORFs of HCMV [6]; out of 54 input ORFs, we identified 29 CCMV homologs. (ii) Our RNA sequencing reads were aligned to the CCMV reference genome; this revealed the presence of 45 splice junctions. By examining these splice junctions for nearby start and stop codons, 28 new ORFs were annotated. (iii) Computational prediction of ORFs was performed using the smORF software [46] and a 6-frame annotation; this added 22 additional ORFs to our list. In total, we found 79 potential ORFs (S2 Table) that were not annotated before [33]. To validate their translation to proteins, we scanned our proteomic data set for matching peptides. In several instances, only peptides were detected that are shared with canonical ORFs and therefore could not be assigned unambiguously. We ended up with 14 newly annotated ORFs, to which at least two unique peptides could be assigned, proving the existence of translation products (Fig 5A and S2 Table).

An example of a newly identified CCMV gene product is UL91.1 that originates from splicing of a 402 codons long alternative reading frame within the UL88 locus to codon 37 of the UL91 ORF, sharing the 107 C-terminal codons with UL91 (Fig 5B). In contrast, the newly identified 413 codons long smORF1, which is encoded within the lytic replication origin (Fig 5C), has no matching sequences in HCMV or other organisms, indicating this ORF to be unique to CCMV. Other newly annotated coding sequences (UL29A, UL70A, UL79A) are partly conserved in HCMV, but the corresponding ORFs are defective and interspersed with stop codons (S9 Fig), indicating that this group of genes was lost during HCMV evolution.

Besides the 14 newly annotated ORFs, we found peptides in our proteomic data set matching N-terminal sequences of UL48A, UL97, UL117 and US19 but spanning over the canonical methionine start codon, indicating N-terminal extensions of the respective gene products. Further, we annotated splicing sites, four long non-coding RNAs and nine micro-RNAs to CCMV BAC-Phan9. An annotated genomic sequence file is available under the NCBI accession number MZ151943.

## Temporal profiling of CCMV protein expression

HCMV gene expression follows a strict temporal order with each protein falling into one of five distinct kinetic classes [8]. To determine whether this level of regulation is conserved in CCMV infection, we determined the kinetic profiles of each identified CCMV protein (n = 142) and clustered these by the k-means method. The optimal number of clusters lies at five for our CCMV data set (S10 Fig). This was in congruence with HCMV [8], indicating that the overall temporal organization of the viral gene expression cascade is conserved between both viruses. Following the nomenclature used for temporal profiles (TP) of HCMV protein

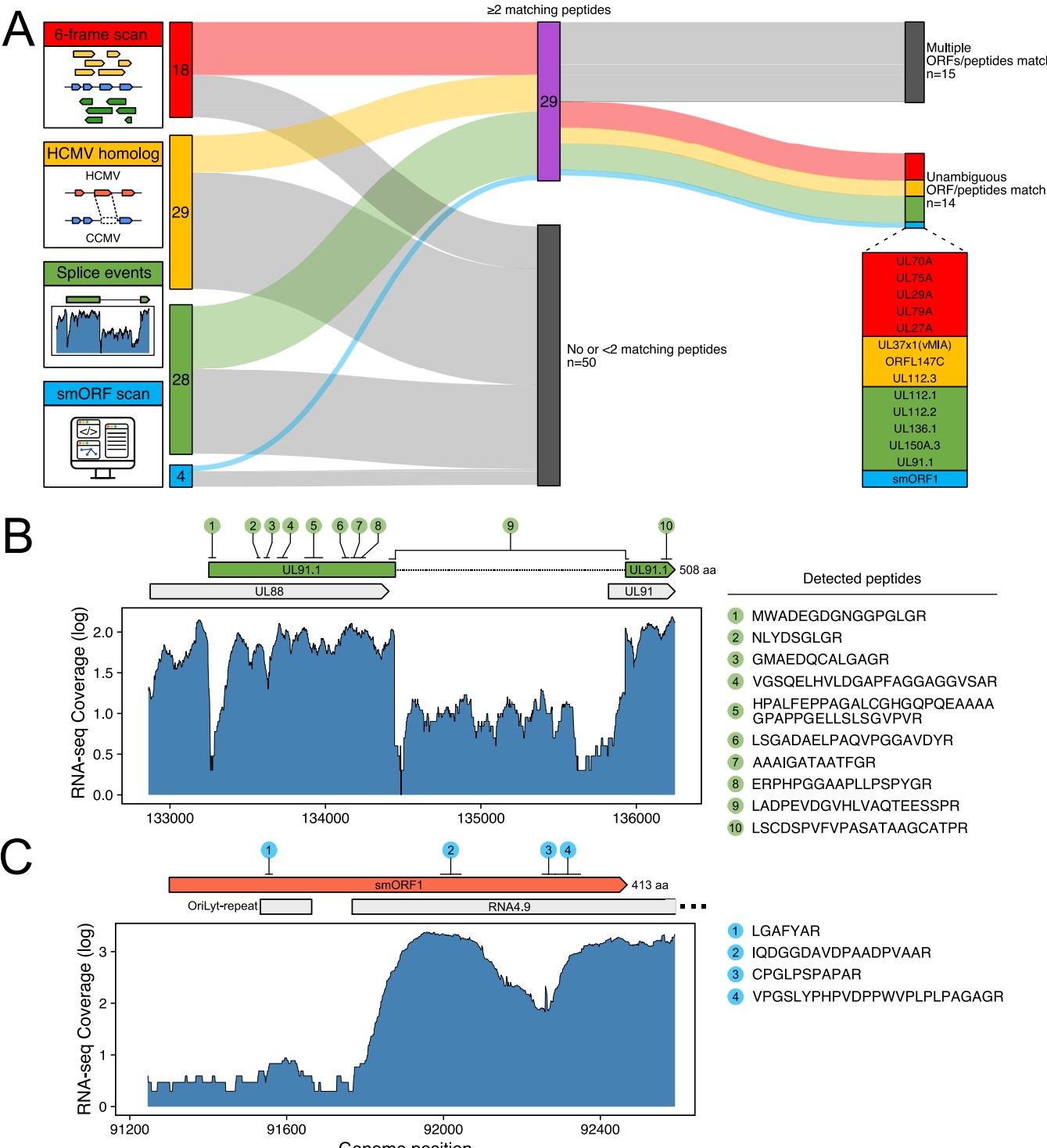

**Fig 5. Re-evaluation of the CCMV coding potential.** (A) To annotate potential novel coding sequences, the CCMV genome was scanned for unknown six-frame translation products, for homologs of non-canonical HCMV ORFs [6], for splice variants and for small ORFs (smORFs). To validate the candidate ORFs (numbers are indicated), proteomic data of CCMV-infected cells was analyzed for matching peptides. In total, 14 novel gene products were unambiguously identified by at least two unique peptides. (B) UL91.1 as an example of a newly found non-canonical CCMV ORF. RNA sequencing (RNA-seq) of infected cells detected a splicing event between the UL88 and UL91 locus. Multiple peptides matched the annotated ORF sequence, including the N-terminal peptide and a peptide spanning over the exon/exon junction, thus confirming translation of this splice variant. (C) smORF1 is another unique ORF with yet unknown function that overlaps the lytic replication origin of CCMV including the 5'-end of the long non-coding RNA4.9.

expression [8], we termed these clusters TP1-TP5 (Fig 6). Similar to HCMV, TP1 and TP4 clusters of CCMV contained the lowest number of proteins and TP5 the highest.

We reasoned that viral proteins that are functionally conserved between HCMV and CCMV should be produced with similar kinetics. Therefore, we compared the individual assignment of homologous viral gene pairs to the temporal clusters (S11 Fig). Genes assigned to the late expression profile (TP5) showed the largest overlap (65% of HCMV, 83% of CCMV proteins) and most non-overlapping genes of TP5 were assigned to the similarly shaped early-late profile TP3 (19% of HCMV, 10% of CCMV TP5 proteins). This indicates that molecular processes driving the accumulation of structural genes are largely conserved between both viruses. In contrast, only one gene of the TP4 clusters overlapped between HCMV and CCMV, suggesting that this small group of genes, with peak expression levels at 48 h post infection, may have less conserved functions.

## Regulation of host gene expression by CCMV

We next aimed to complete our gene expression analysis of CCMV infected cells by evaluating the level of host gene regulation. In total, we quantified 4,748 host proteins with at least two matching peptide species (S5 Table). To allow direct comparison to HCMV, we included in our analysis a previously published dataset of HCMV-infected cells [47]. First, we calculated the changes in protein abundances induced by CCMV and HCMV and aligned the datasets based on shared Uniprot IDs (S6 Table). Then, we determined the top 3% most up- and down-regulated host proteins at each time point (194 proteins) of CCMV and HCMV infection (Fig 7A). To identify functionally related proteins enriched in the individual gene sets, we conducted a gene ontology (GO) analysis [48]. At late times of infection, both viruses showed a strikingly similar pattern in the down-regulation of pathways associated with actin cytoskeleton and extracellular matrix organization, cell adhesion, Rho GTPase, growth factor and cytokine signaling as well as programmed cell death (Fig 7B, lower panel), suggesting that previously reported HCMV gene functions [20,49–52] are conserved in CCMV. At early times of infection, HCMV and, to a lesser extent, CCMV interfered with the expression of protein clusters associated with dNTP and amino acid metabolism as well as vesicle-mediated transport. Conversely, both viruses showed an early induction of proteins involved in mRNA processing and DNA repair (Fig 7B, upper panel). Moreover, protein folding and a broad spectrum of metabolic processes were similarly enriched by HCMV and CCMV at later stages of infection. Of note, several GO terms describing mitochondrial processes, known to be important for HCMV infection [53], are up-regulated solely by HCMV.

Next, we characterized the dynamics of mRNA and protein expression of 4,748 host genes during CCMV infection in order to distinguish between transcriptionally regulated host factors and those that are down-regulated at the protein level. We performed a k-means clustering analysis (S13 Fig) and found that the host gene expression profiles fall into seven distinct groups (Fig 8). Exactly the same optimal number of clusters was found by Nightingale et al. when analyzing host mRNA and protein expression patterns in HCMV-infected fibroblasts [47]. Three clusters in our analysis show relatively stable transcript levels but decreased protein amounts (Fig 8, left panel) at either early (cluster 6), late times (cluster 5) or throughout infection (cluster 7). Such patterns are typical for virus-induced degradation of proteins hindering the infectious process. In fact, clusters 5 and 7 include the cyclin-dependent kinase inhibitor CDKN1A and the E3 ubiquitin ligase NEDD4 (Fig 8, right panel), which are known targets of protein degradation by HCMV [47,54]. Cluster 6 includes MYO18A which shows an initial degradation but a rapid increase at later stages of infection, which is in line with MYO18A being repurposed for virus assembly and egress during HCMV infection [55].

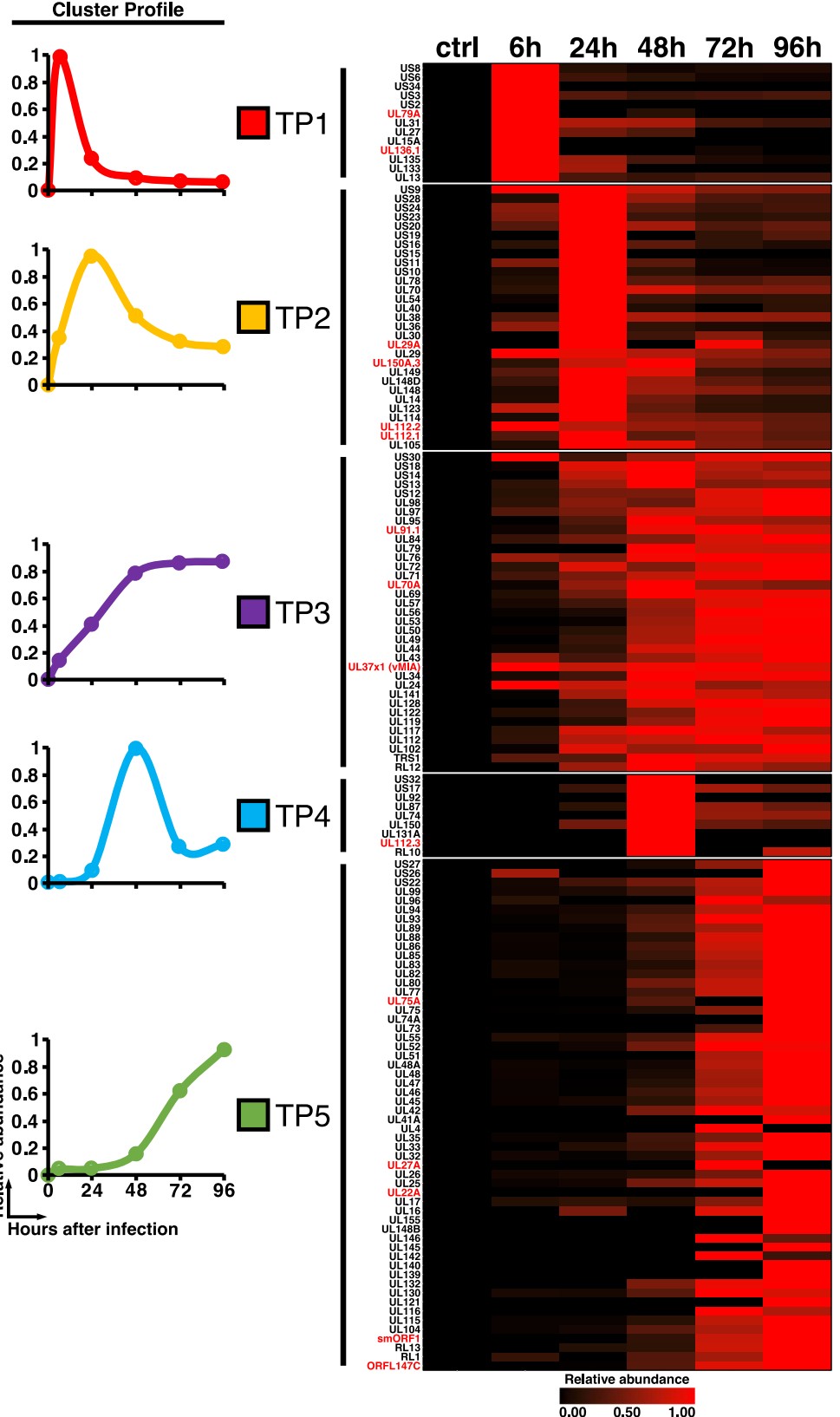

**Fig 6. Temporal classes of CCMV protein expression.** CCMV-infected cells were harvested at the indicated times post infection and subjected to a mass spectrometry-based proteomic analysis. K-means clustering classified viral gene expression into five temporal profiles (TP). The left-hand panel displays the average cluster profiles of TP classes 1–5. On the right hand, the expression profiles of individual CCMV proteins are plotted as a heatmap.

Two other clusters contain genes whose protein levels closely parallel mRNA expression, indicating transcriptional activation (cluster 2) and repression (cluster 4), respectively. 49% of transcriptionally induced and 56% of transcriptionally silenced genes in CCMV infection are also present in the corresponding clusters from HCMV-infected cells (S14 Fig). This suggests that the regulation of host gene transcription is generally well conserved between both viruses. An example of a transcriptionally repressed gene during CCMV infection is the dNTP hydrolase SAMHD1 which acts as a restriction factor for many viruses including HCMV [56,57]. Thus, our database of mRNA and protein profiles in CCMV infection can function as a valuable resource for identifying host targeting mechanisms shared with HCMV and sets the framework for a comparative assessment of conserved virus-host interactions.

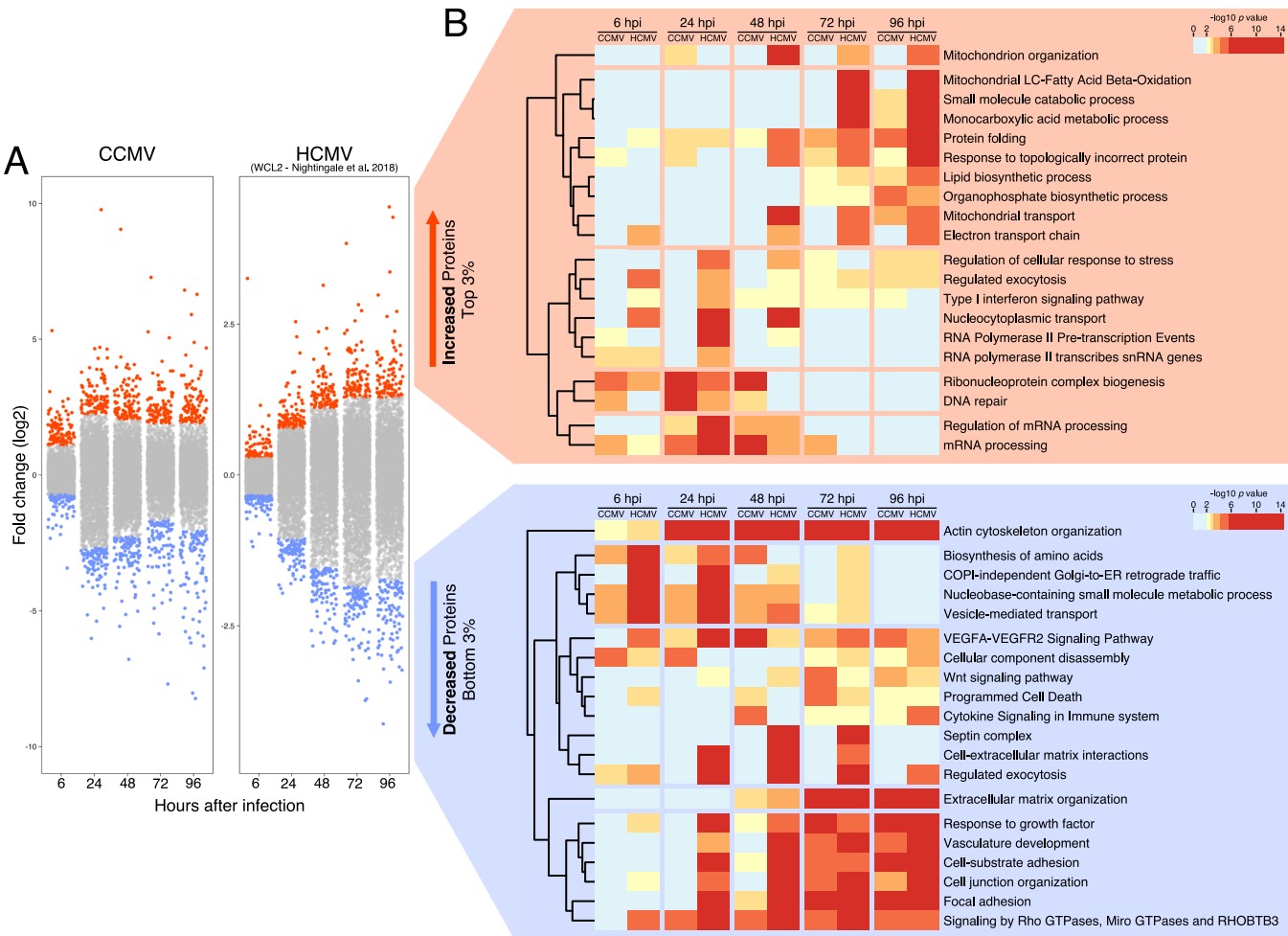

**Fig 7. Gene ontology (GO) analysis of the most deregulated host proteins during CCMV and HCMV infection.** (A) For each time point post infection, fold-changes in host protein expression were calculated relative to uninfected control cells (0 h). The top and bottom 3% of protein fold-changes were selected from each time point and used for multi-list GO enrichment analysis. (B) Heatmap showing the GO terms for strongly increased or depleted proteins during HCMV and CCMV infection (LC: long chain). The HCMV data set WCL2 was taken from [47].

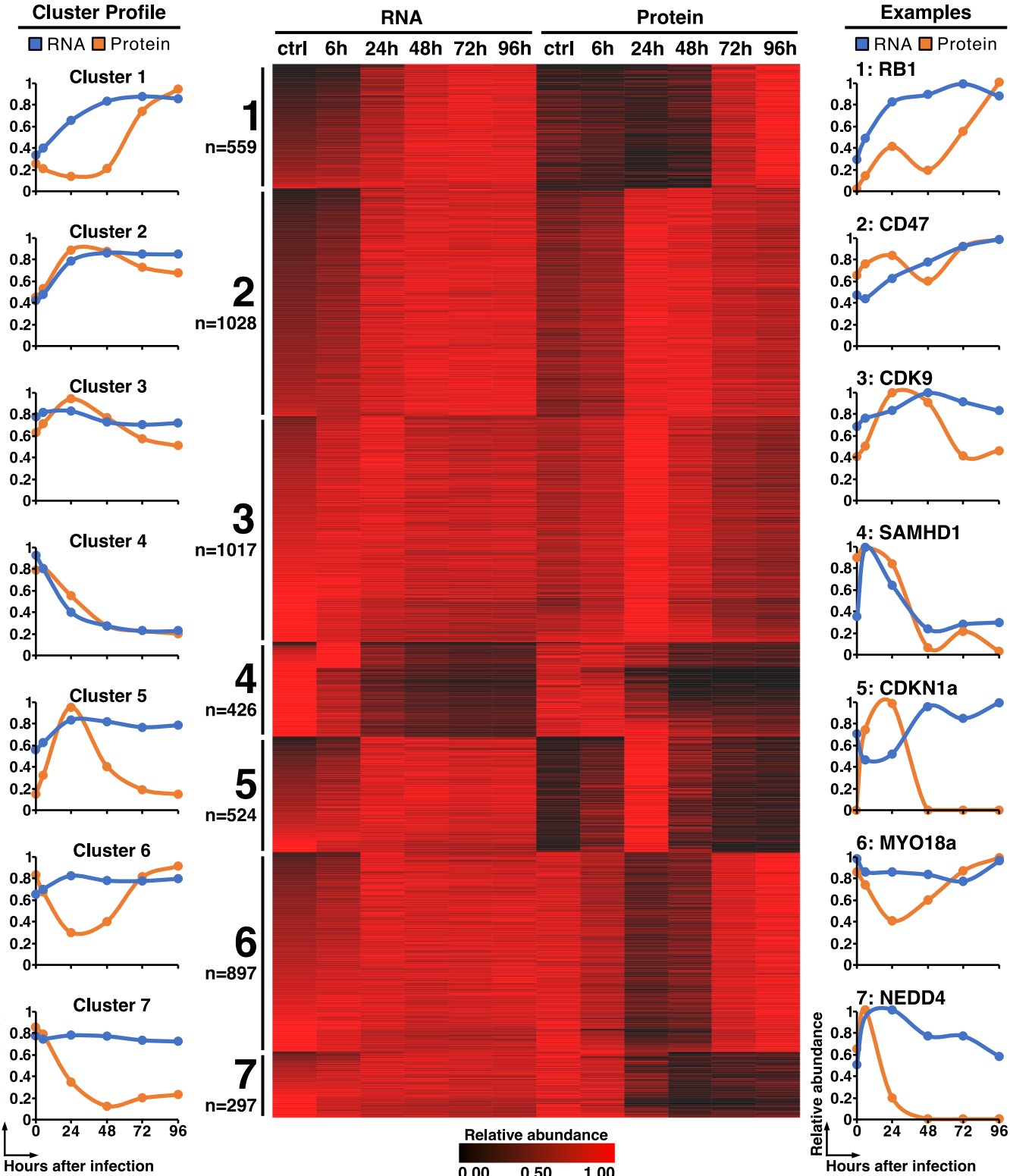

**Fig 8. Comparative analysis of CCMV-induced changes in host gene expression at the transcript and protein level.** At the indicated time points total RNA and protein were prepared from CCMV- infected cells and analyzed by RNA sequencing and tandem mass spectrometry respectively. K-means clustering of 4,748 transcripts and proteins identifies seven individual clusters representing different modes of host gene regulation during CCMV infection. Shown are the average mRNA and protein profiles of each cluster (left panel), a heatmap of the corresponding gene expression data (middle panel) and selected representative profiles from each cluster (right panel).

## Discussion

Animal CMVs and their BAC clones have been utilized for decades for *in vivo* and *in vitro* models and thereby contributed immensely to the understanding of pathogenesis, host immune response and underlying molecular mechanisms of CMV infections. As the closest relative to HCMV, the chimpanzee CMV provides a link between the human and other animal CMV species and holds a unique potential to serve as a model system for HCMV infection. Here, we established a BAC tool based on the genome of the CCMV Heberling strain, making CCMV for the first time readily accessible to reverse genetic approaches. The systems-level characterization of RNA and protein expression led to the identification of novel viral gene products and highlighted the many shared attributes of CCMV and HCMV gene regulation, providing resource and guidance for the usage of CCMV as a model.

The straightforward approach to integrate the BAC cassette into the viral genome by homologous recombination proved to be a reliable cloning strategy for CCMV as it has been for HCMV genomes before [22,34,35]. However, recombinant viral genomes exceeding the natural genome length provoke deletions or genomic rearrangements to accommodate the viral genome into the capsid structure. Indeed, deletions of in total 8.8 kbp adjacent to the BAC cassette were detected in all our initially selected BAC clones (Fig 1) and similar deletion events occurred during the BAC-cloning of HCMV [36,37]. While this sets a limitation on BAC-capturing the full length viral genome at first instance, deleted sequences can be seamlessly reinstated later by BAC mutagenesis and by making the BAC cassette excisable as it has been done in this study and for many HCMV strains before [36,58,59]. In the future, approaches based on the assembly of synthetic or separately cloned subgenomic fragments may reduce the risk of such unwanted deletion and recombination events and become attractive alternatives to traditional BAC cloning procedures [60–62].

Whole genome sequencing of the Heberling strain revealed two polymorphic loci, US26-28 and UL48-55, within the CCMV genome and a large set of the variations carried over in different combinations to the five sequenced BAC clones (Figs 2A and S6). There are several possible explanations for the origin of this genetic diversity. As CCMV Heberling was never plaque-purified after isolation from the donor animal, some of the observed variations may reflect the pre-existing viral intra-host diversity. For instance, the UL55 gene of HCMV, encoding glycoprotein B (gB), is also polymorphic and occurs in mixed genotypes in infected individuals [63]. The positions of non-synonymous mutations found in CCMV-UL55 directly correspond to the most variable gB regions in HCMV (S15A Fig), including antigenic domains 1–2 and the furin cleavage site [64,65]. This suggests that the different UL55 genotypes of CCMV represent neutralizing antibody escape variants originating from the original chimpanzee sample.

Another source of genetic diversity are evolutionary processes like selection and genetic drift, acting on CCMV during *in vitro* culture. The CCMV isolate not only had to adapt to a human host cell environment but was selected for at least twenty passages for cell-free growth on fibroblasts [33], without any selective pressure by the immune system. Well known adaptations of HCMV to the cultivation on fibroblasts are disruptive mutations in the UL128 locus, the RL13 gene and the ULb' region [23]. While CCMV Heberling contains a frameshift mutation in UL128, facilitating the growth on fibroblasts but rendering the virus unable to infect endothelial and epithelial cells (Fig 3), RL13 and ULb' genes lack any obvious frameshift or translation terminating mutations. Instead, we found that a genomic deletion ranging from IRS1 to US10 has been accumulated within the Heberling stock. This is in analogy to the loss of the ULb' region which is frequently observed in HCMV laboratory strains upon extended passage on fibroblasts [66]. Whether the IRS1-US10 deletion provides a size-based replication advantage or eliminates gene functions that slow down CCMV growth on fibroblasts, or

human cells in general, needs further investigation. This is facilitated with our BAC-tool as we partially repaired this region by inserting the US2-US10 gene locus. As a consequence, the BAC-derived virus can express all genes from the US2-US10 locus to higher levels than the parental Heberling stock (Fig 4C) and is able to down-modulate MHC-I expression at the cell surface (Fig 4D). Thus, fixing the CCMV sequence by BAC-cloning allows robust experimentation.

Adaptation processes of primate CMVs to human cells can occur in a rapid fashion facilitated by recombination events [3] or by positive selection of advantageous genetic variants [1]. UL48, which exhibits the second highest number of non-synonymous polymorphisms among all CCMV genes (Fig 2B and S1 Table), has been predicted to be one of the factors driving the evolutionary adaptation to the human host [1]. The fact that one half of all non-synonymous polymorphisms in CCMV-UL48 cluster to a narrow central region whereas none is located in the N-terminal deubiquitinase domain (S15B Fig), argues that non-enzymatic functions of this inner tegument protein might be involved in such adaptation processes [67,68]. In contrast to the UL48-UL55 region, which encompasses essential core genes, the overall high frequency of nucleotide variants across the US region (Figs 2B and S6) is likely due to the fact that US genes are dispensable for virus growth *in vitro* [13–15] and therefore tolerate mutations.

It is yet unclear to what extent our BAC-Phan9 clone represents an authentic wild-type CCMV. This is due to the fact that the Heberling isolate has genetically adapted to human fibroblasts before sequencing [33] and that CCMV sequence information obtained from chimpanzee specimens is sparse [2]. One option to address this question is to evaluate whether CCMV BAC-Phan9 and the Heberling strain have retained the ability to grow on chimpanzee host cells. Moreover, a systematic comparison of CCMV infection in human versus chimpanzee cells can shed light on whether there is a species barrier and its molecular determinants. It is thus important to interpret our data in the context of an experimental system that does not fully recapitulate infection in the natural host.

By re-evaluating the coding potential of CCMV we identified 14 novel ORFs that could be confirmed by mass spectrometry (Fig 5A and S2 Table). Among the newly annotated genes, several are unique to CCMV, including the UL91 splice variant UL91.1 and the oriLyt-spanning smORF1 (Fig 5B and 5C). UL91 is an integral part of the viral pre-initiation complex (vPIC) that governs the transcriptional activation of "true" late genes [69]. The UL91 N-terminus (residues 1–70) is conserved among primate CMVs and its integrity is essential for vPIC function [70]. Considering that residues 1–36 of UL91 are lacking in UL91.1, it is highly unlikely that the UL91 activity is preserved. Possibly, the shared C-terminus enables UL91.1 to act as a regulator of vPIC, either by competing with UL91 binding or by recruitment of the large, intrinsically disordered N-terminal domain encoded by UL91.1 exon 1. Although UL91.1 splice sites are conserved in RhCMV and HCMV [7], a UL91.1 exon 1-related ORF is completely lacking in these viruses. This stands in contrast to UL29A, UL70A and UL79A which, although being unique to CCMV, have homologous, yet fragmented and non-functional ORFs at related positions of the HCMV genome (S9 Fig). Hence, it appears that UL91.1 has been newly acquired by CCMV while UL29A, UL70A and UL79A derive from common ancestors that were lost during HCMV evolution.

The finding that smORF1 overlaps with the oriLyt of CCMV was surprising, as lytic replication origins of herpesviruses are typically devoid of protein coding sequences and instead regulated by non-coding transcription [71]. The long-noncoding RNA4.9 of HCMV activates viral DNA replication by R-loop formation, which exposes the oriLyt to the viral single-stranded DNA-binding protein UL57 [72]. The abundant expression of an RNA4.9 homolog from the oriLyt region of CCMV (Fig 5C) indicates that this mechanism is conserved, despite the presence of smORF1. Concerning the potential conflicts between transcription and replication processes [73], the

kinetics of smORF1 expression (TP5, Fig 6 and S4 Table) may indicate that smORF1 transcription impacts the initiation of viral DNA synthesis at late stages of CCMV infection.

The temporal analysis of CCMV protein expression reveals that the majority of gene products fall in the same kinetic classes as their HCMV orthologues (S11A Fig) and most of the observed class switches can be considered as conservative (S11B Fig). Nevertheless, a few orthologs show drastic differences in their kinetic profiles (S12 Fig). This latter group includes UL148 which has early expression kinetics in CCMV (TP2) but late kinetics (TP5) in HCMV infection. HCMV-UL148 has two distinct functions: it promotes the intracellular retention of CD58, thereby impeding the detection of infected cells by the immune system [74]; and it causes large-scale remodeling of the endoplasmic reticulum (ER), thereby protecting the viral glycoprotein O (UL74) from ER-associated degradation [75,76]. Since only the immune evasion but not the ER remodeling function is conserved in CCMV-UL148 [76], it is conceivable that its earlier expression reflects the lack of ER remodeling function. In fact, HCMV-UL74 expression (TP5) parallels that of UL148 (S11 and S12 Figs), suggesting that the temporal shift of HCMV-UL148 expression has evolved to ensure proper maturation of viral envelope proteins during the assembly phase of infection. This would support a general scenario, where distinct expression kinetics indicate functional divergence of viral orthologs.

Our systems-level overview of host dynamics during CCMV infection and comparison to HCMV data from previous studies highlights many common themes of CMV infection (Fig 7B) and thus further underscores the modelling potential of CCMV. For example, clusters 5 and 7 contain host factors with steady mRNA levels but a rapid decrease in protein abundances as infection progresses (Fig 8), suggesting these factors are targets of virus-induced protein degradation. Clusters 5 and 7 include NEDD4, NEDD4L and ITCH, members of the NEDD4 family of E3 ubiquitin ligases that recognize a PPXY docking motif on their protein substrates [77]. HCMV-UL42 employs two PPXY motifs in its N-terminus to interact with NEDD4 family members and causes their proteolysis [11,78,79], possibly to protect other viral proteins from NEDD4-mediated ubiquitination [80]. The presence of three PPXY motifs in the CCMV-UL42 N-terminus supports the notion that NEDD4 degradation is a shared strategy of herpesviruses [78].

Our searchable database with transcript and protein dynamics of 4,748 cellular genes allows a quick assessment of further relevant virus-regulated host factors and signaling pathways (S5 Table). This can provide insights into host subversion strategies that are shared between CCMV and HCMV. Such comparative analyses are instrumental for the identification of critical nodes of virus-host interaction. In addition, the CCMV system offers a versatile genetic tool to dissect essential functions and infection mechanisms of HCMV by gene or domain swap experiments. All these comparative approaches are greatly facilitated by the fact that both viruses can be analyzed in the same cellular background. Thus, although CCMV research is limited to *in vitro* systems [81], the close phylogenetic relationship of CCMV to HCMV offers some key advantages over other CMV species that, with the help of our in-depth characterized virus clone, can be exploited in future studies.

## Materials and methods

### Cell culture and viruses

Human embryonic lung fibroblasts (Fi301 –obtained from the Institute of Virology, Charité) were maintained as previously described [82]. HUVEC cells were kindly provided by Andrea Weller (Center for Cardiovascular Research, Charité) and maintained as previously described [83]. ARPE-19 cells (ATCC CRL-2302) were cultured in DMEM 10% FCS with 20 mM L-Glutamine and 45 μg/ml gentamicin. The CCMV Heberling strain was kindly provided by Gary S. Hayward (Johns Hopkins School of Medicine, Baltimore, MD, USA). For preparation of viral

stocks, CCMV Heberling strain was added to Fi301 cells at an MOI of 0.1 and infected cells were kept in subculture until all cells showed a cytopathic effect. The medium was then changed to DMEM supplemented with 2% FCS, 20 mM L-Glutamine and 45 μg/ml Gentamicin (harvest medium). Viral supernatant was harvested 3–4 days post medium change, centrifuged at 3,000 rcf for 15 minutes and stored as 2 ml single-use aliquots at -80°C. For infection experiments that required a high MOI, virus was concentrated prior to infection by ultracentrifugation at 30,000 rcf for 40 minutes at 25°C on a 20% sucrose cushion. The virus was then resuspended in medium and added to the cells at the desired MOI.

## BAC cloning

he BAC donor plasmid pEB1097 (kindly provided by Eva Borst, Hannover) contains a BAC cassette flanked by homology arms suited to the US1-2 and US6-7 sequences of HCMV strain AD169 [35]. The homology arms of pEB1097 were replaced with the corresponding genomic regions of CCMV Heberling (coordinates left homology arm: 204,201–205,187 bp, right homology arm: 208,104–209,072 bp) via NEBuilder HiFi DNA Assembly Cloning (New England Biolabs). The resulting vector was named pEB1097_CCMV (S1 Fig). Recombinant virus was generated by spontaneous recombination of the BAC cassette and the CCMV genome in infected cells. First, the BAC cassette including the homology arms was PCR-amplified from pEB1097_CCMV, using the primer pair 69_CCMV_donor_PCR_fw and 70_CCMV_donor_PCR_rev. 1 μg of the amplicon was transfected into $10^6$ Fi301 cells by Amaxa nucleofection, using the "Primary Fibroblast" kit and pulse program A-023 (Lonza). The day after, cells were infected with CCMV Heberling at an MOI of 1. 4 days post infection, virus containing supernatant was transferred to a fresh culture of Fi301 cells. From then on, selection was applied by adding 200 μM xanthine and 200 μM mycophenolic acid to the culture medium. After three rounds of virus passages with selection, the circular virus DNA was extracted from infected cells using the Hirt method [84] and electroporated into *E.coli* Electro-MAX DH10B Cells (Invitrogen). Transformants were selected on LB agar plates with 12.5 μg/ml chloramphenicol and colonies were picked for further characterization.

## BAC mutagenesis

CCMV BAC mutants were created by en passant mutagenesis as previously described [85]. Authentic recombination events were verified via PCR analysis and Sanger sequencing. Overviews of the BAC cloning strategies are shown in S2–S4 Figs and S3 Table lists all oligonucleotides used in this study.

## Reconstitution of CCMV BAC virus

BAC-DNA was prepared using the NucleoBond Xtra-Midi kit (Macherey-Nagel) according to the manufacturer's instructions. A mixture of 2 μg of BAC-DNA, 2 μg pcDNA-pp71-flag and 1 μg of pBRep-Cre (obtained from Wolfram Brune, Hamburg, Germany) was electroporated into Fi301 cells using the Amaxa nucleofector program A-23 (Lonza). Plaque formation was checked over the course of 14 days, cells were kept in subculture and once all cells showed a cytopathic effect, medium was changed to harvest medium and viral supernatant was harvested 2–3 days post medium change.

## Viral DNA extraction

For isolation of viral DNA, viral particles were ultra-centrifuged at 30,000 rcf for 30 minutes at 25°C. Supernatant was discharged and the pellet was resuspended in 100 μl Tris buffer

containing proteinase K and incubated overnight at 56˚C. DNA was then extracted via Phenol-Chloroform extraction using Roti-Phenol (Carl Roth). The formation of genomic isoforms was analyzed by PCR and Sanger sequencing.

## Flow cytometry

CCMV infectious titers were assessed based on the ability to induce IE1 expression 24 hours after infection. Infected cells were harvested by trypsinization, fixed and permeabilized by ice-cold absolute ethanol and incubated for at least 10 minutes at 4˚C. Cells were then stained with Alexa Fluor 488-conjugated anti-IE1/IE2 (MAB810X) antibody overnight at 4˚C. MHC-I surface expression was analyzed as described elsewhere [36]. Cells were harvested by incubation in 0.04% EDTA containing phosphate-buffered saline (PBS) and stained using a phycoerythrin (PE)-conjugated anti-HLA-ABC antibody (clone W6/32) and an appropriate PE-conjugated mouse IgG2a κ isotype control (clone eBM2a). Both antibodies were obtained from Thermo Fisher Scientific. Flow cytometry was done on a FACSCanto II flow cytometer (BD Biosciences) using FACSDiva (BD Biosciences) and FlowJo (FlowJo LLC) software packages. Cellular debris, cell doublets and aggregates were gated out of analysis.

## CCMV DNA sequencing and variant analysis

DNA from CCMV BAC clones and the CCMV Heberling stock was sequenced at Eurofins Genomics with Illumina 2x150 bp paired end-reads. Eurofins variant analysis included read-mapping against the given reference genome, detection and annotation of single nucleotide variations (SNVs) and InDels as well as the allocation of their effects on protein level. CCMV genome region, ORFs, DNA sequencing coverage and positions of SNVs were plotted using the Circos 0.69–9 software package.

## Total RNA sequencing

Confluent Fi301 cells were infected in triplicates with CCMV BAC-Phan9 using an MOI of 5. Samples were harvested by resuspension in Trizol reagent. RNA extraction, library preparation, total RNA sequencing and data processing were performed as previously described [10]. Splice junctions were identified using Geneious Prime software.

## Mass spectrometry

Infection conditions were the same as for the sample preparation for RNAseq. Infected cells were harvested by removing the medium and scraping off the cell layer in 1 ml PBS. For lysis, the cell pellet was resuspended in 200 µl of lysis buffer: 7 M Urea, 50 mM triethylammonium bicarbonate (pH 8.5), 1% Triton-X-100, 5 mM Tris(2-carboxyethyl)phosphine, 30 mM chloroacetamide, complete mini EDTA free protease inhibitors (Roche), 2 µl Benzonase (Merck-Millipore) and sonicated for 45 min (intervals: 20 s on, 40 s off) using a Bioruptor (Diagenode). After clearing the lysates, protein was isolated using methanol/ chloroform precipitation, according to standard protocols [86]. The precipitates were resuspended in digestion buffer, consisting of 50 mM triethylammonium bicarbonate (pH 8.5), 1% sodium deoxycholate, 5 mM Tris(2-carboxyethyl)phosphine, 30 mM chloroacetamide, Trypsin (1:25, w/w), Lysyl Endopeptidase (1:100, w/w), and digested overnight at room temperature. Peptides were desalted using two disks of C18 material embedded in Stage Tips [87] and stored at 4˚C until LC-MS/MS measurement.

The LC/MS analysis was performed using a Thermo Scientific Dionex UltiMate 3000 system connected to a µPAC trapping column (PharmaFluidics). The mobile phase A contained

2% acetonitrile and 0.05% Trifluoroacetic acid (TFA) in water and the mobile phase B contained 0.05% TFA in acetonitrile. For peptide separation a 200 cm μPAC micro-pillar array column at 180 min gradient length was employed, with mobile phase A containing 0.1% formic acid in water, and mobile phase B containing 0.1% formic acid in acetonitrile. The flow rate was set to 750 nL/min from 4% to 20% B and 350 nL/min from 20% to 80% B.

The MS1 scans were performed in the orbitrap using 120,000 resolution. Peptides were fragmented using higher-energy collision induced dissociation (HCD) with 30% HCD collision energy. The MS2 scans were acquired in the ion trap with standard AGC target settings, an intensity threshold of 5e3 and maximum injection time of 40 ms. A 1 s cycle time was set between master scans.

Raw-files were searched using MaxQuant version 1.6.2.6 [88]. Search parameters included two missed cleavage sites, fixed cysteine carbamidomethyl modification, and the variable modifications methionine oxidation, N-terminal protein acetylation as well as asparagine–glutamine deamidation. The "match between runs", "iBAQ" (intensity-based absolute quantification), "second peptide" and "LFQ" (label-free quantification) options were enabled [89]. Database search was performed using Andromeda, the integrated MaxQuant search engine, against a Uniprot database of homo sapiens proteins (downloaded 2020), the predicted amino acid sequences of the CCMV BAC and the proteins from a six frame translation of the CCMV genome (minimum sequence length: 15 amino acids) with common contaminants. False discovery rate was estimated based on target-decoy competition using a reverted database and was set to 1% at peptide spectrum match, protein and modification site level.

## Data analysis

For the multi-set gene ontology-enrichment analysis of the top 3% of proteins (highest and lowest log2 fold-change), the Metascape tool was applied [48]. Protein level data was matched to RNA level data based on the HGNC official gene symbol and were clustered using the k-means function in R [90]. We determined the optimal number of clusters by calculating the summed distance of each protein from its cluster centroid and plotted these values against the corresponding numbers of clusters. The inflection point of this curve indicates the optimal number of clusters ("elbow method"). Plots were created using ggplot2 [91] and pheatmap v.1.0.12 (https://CRAN.R-project.org/package=pheatmap).

## Supporting information

**S1 Table. Variant analysis of CCMV BAC clones and the parental CCMV Heberling stock.** Variant analysis of CCMV BAC-165 (sheet 1), BAC-169 (sheet 2), BAC-177 (sheet 3), BAC-185 (sheet 4), BAC-208 (sheet 5), Heberling stock (sheet 6) and BAC-Phan9 (sheet 7). BAC variations not found in the CCMV Heberling stock are marked yellow.
(XLSX)

**S2 Table. Newly identified ORFs and regulatory RNAs of CCMV.** Novel ORFs found during the coding potential evaluation of CCMV are listed in sheet 1. CCMV regulatory RNAs annotated based on sequence homology to HCMV are listed in sheet 2.
(XLSX)

**S3 Table. List of oligonucleotides used in this study.**
(XLSX)

**S4 Table. Relative abundances and temporal profile classification of CCMV proteins.** LFQ values were the basis for the calculation of expression profiles. The maximum LFQ for each gene across time points was set to 1. The remaining time points were calculated relative to this

maximum. A value of 0 indicates that the respective protein could not be quantified at the respective time point.
(XLSX)

**S5 Table. Relative abundances of host RNA and protein during CCMV infection.** Relative abundances and temporal profile classification of host proteins during CCMV infection (sheet 1) and HCMV infection (sheet 2). LFQ values (CCMV) or TMT-based values (HCMV) were the basis for the calculation of expression profiles. The maximum values for each gene across time points was set to 1. The remaining time points were calculated relative to this maximum. A value of 0 indicates that the respective protein could not be quantified at the respective time point. The HCMV data set (sheet 2) was taken from S1 Table in [47].
(XLSX)

**S6 Table. Log2-fold changes of host proteins during CCMV and HCMV infection.** For host proteins during CCMV infection (sheet 1), calculations were done by log2 transformation of LFQ values, comparing the indicated time points to time point 0 h. For HCMV infection (sheet 2), log2-fold changes were calculated from dataset WCL2 [47], comparing the indicated WCL2 infection time points to the average of WCL2-Mock1 and WCL2-Mock2. Only proteins present in both the CCMV and the HCMV dataset are included.
(XLSX)

**S1 Fig. Cloning scheme of the BAC donor vector pEB1097_CCMV.** Homology arms for targeted integration into CCMV genome region US2-US6, BAC cassette and plasmid backbone sequences were PCR amplified and assembled via Gibson assembly. Plotted with SnapGene 6.
(EPS)

**S2 Fig. Cloning of the transfer vector for the restoration of the deleted sequences on the left hand side of the BAC cassette in CCMV BAC 177.** Plotted with SnapGene 6.
(EPS)

**S3 Fig. Cloning of the transfer vector for the restoration of deleted sequences on the right hand side of the BAC cassette in CCMV BAC 177.** Restoration was done in two parts: (A) Region US8-US10 (B) Region US11-US13. Plotted with SnapGene 6.
(EPS)

**S4 Fig. PCR amplification scheme of the *en passant* recombination template for the UL128 frameshift repair.** Plotted with SnapGene 6.
(EPS)

**S5 Fig. Deletion events in the selected CCMV BAC clones.** (A) PCR spanning over the left hand side of the BAC cassette integration site reveals deletion of IRS1 to US1 (3,639 bp). (B) PCR at the right hand side of the BAC integration site reveals deletion of US8 to US13 (5,191 bp) in all clones. (C) Coverage plots show alignment of sequencing reads to adjusted reference genome. Alignment confirms deletion events and also reveals the same deletions in clones 165, 169 and 208 at the left hand side of the BAC cassette. Tracks in the inner circles show the average coverage in 500 bp increments (blue) of each clone, CCMV ORFs (red) and genomic regions (white) are plotted in the outer tracks of the Circos plot.
(EPS)

**S6 Fig. Distribution of silent and missense mutations in selected CCMV BAC clones and the parental Heberling strain.** Silent mutations are clustered at two polymorphic loci with a higher prevalence than missense mutations.
(EPS)

**S7 Fig. CCMV BAC-Phan9 forms all four genomic isoforms.** (A) Virion DNA from Heberling and BAC-Phan9 virus served as PCR template in order to identify the four viral genomic isoforms. Expected amplicon sizes are indicated. The correct gel bands for Heberling isoforms 1 and 2 are weak, reflecting the fact that an IRS1-US10 genomic deletion prevails in the Heberling stock. (B) Large arrows indicate the orientation of the US and UL region in each isoform. Small arrows indicate the PCR layout and primer names. Sanger sequencing of PCR amplicons confirmed the recombination events and the presence of all four isoforms in BAC-Phan9 virions despite the deletion event in the internal repeat region in its parental BAC vector.
(EPS)

**S8 Fig. Transcriptional landscape of US2-9 region at 6 hpi.** RNA sequencing coverage indicates seperate transcriptional units US2, US3-US6 and US8-US9.
(EPS)

**S9 Fig. Sequence regions in HCMV Merlin corresponding to the newly found CCMV ORFs.** tBlastn scan identified sequence regions in HCMV Merlin (blue marked boxes) homologous to UL29A, UL70A and UL79A of CCMV. Genome coordinates, conserved splice sites, stop codons arising from frame-shift mutations and the percentage of amino acid identities and positives (amino acids that are either identical between the query and the subject sequence or have similar chemical properties) are indicated for the aligned sequence regions.
(EPS)

**S10 Fig. Number of temporal classes of CCMV gene expression.** The summed distance of each protein from its cluster centroid was calculated for one to 10 clusters and plotted. The point of inflection fell at five.
(EPS)

**S11 Fig. Temporal profile (TP) class comparison between HCMV and CCMV.** (A) Venn diagram shows the amount of overlap of homologs in each TP class. (B) Sankey diagram shows the TP class switch of homologs from CCMV to HCMV.
(EPS)

**S12 Fig. Comparison of CCMV and HCMV protein expression kinetics.** The temporal expression profiles of CCMV and HCMV protein homologs were compared side by side on a relative scale. HCMV data were taken from [8].
(EPS)

**S13 Fig. Determining the number of temporal classes of host gene expression during CCMV infection.** The summed distance of each gene from its cluster centroid was calculated for one to 20 clusters and plotted. The point of inflection fell at seven.
(EPS)

**S14 Fig. Cellular factors with similar expression profiles during HCMV and CCMV infection.** Selected clusters with kinetics that indicate biological relevant modes of regulation (Transcriptional up- and down-regulation, protein degradation) were compared between CCMV and HCMV data sets [47]. Left and right panels show the average cluster profiles. Venn diagrams show the overlap of genes in paired clusters.
(EPS)

**S15 Fig. Nature and position of amino acid (aa) changes in UL48 and UL55 (gB).** (A) Mutation clusters found in UL55 (gB—922 aa) of CCMV 169 with corresponding sequence areas of antigenic domains (AD) and variable regions (VR) described in HCMV-gB (902 aa). Positions

of HCMV-gB domains taken from [65]—HCMV Envelope Glycoprotein Diversity Demystified. (B) Amino acid changes in UL48 (2276 aa) of CCMV BAC 165. The corresponding sequence region of the HCMV-UL48 DUB domain is indicated.
(EPS)

## Acknowledgments

We are grateful to Benedikt Kaufer (Free University Berlin) for his helpful advice regarding the BAC cloning procedure. We thank Andrea Weller (Charité) and Eva Maria Borst (Medizinische Hochschule Hannover) for providing valuable reagents. We also thank Iris Gruska and Barbara Vetter for their excellent technical assistance.

## Author Contributions

**Conceptualization:** Quang Vinh Phan, Boris Bogdanow, Lüder Wiebusch.

**Data curation:** Quang Vinh Phan, Boris Bogdanow, Emanuel Wyler.

**Formal analysis:** Quang Vinh Phan, Boris Bogdanow, Emanuel Wyler.

**Funding acquisition:** Quang Vinh Phan, Boris Bogdanow, Christian Hagemeier, Lüder Wiebusch.

**Investigation:** Quang Vinh Phan, Boris Bogdanow, Emanuel Wyler, Lüder Wiebusch.

**Methodology:** Quang Vinh Phan, Boris Bogdanow, Emanuel Wyler.

**Project administration:** Christian Hagemeier, Lüder Wiebusch.

**Resources:** Quang Vinh Phan, Markus Landthaler, Fan Liu, Lüder Wiebusch.

**Supervision:** Boris Bogdanow, Christian Hagemeier, Lüder Wiebusch.

**Visualization:** Quang Vinh Phan.

**Writing – original draft:** Quang Vinh Phan, Boris Bogdanow, Lüder Wiebusch.

**Writing – review & editing:** Quang Vinh Phan, Boris Bogdanow, Lüder Wiebusch.

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
