## [Decision Letter · Decision Letter 0]

30 Aug 2021

Dear Dr. Wiebusch,

Thank you very much for submitting your manuscript "Engineering, decoding and systems-level characterization of chimpanzee cytomegalovirus" for consideration at PLOS Pathogens. As with all papers reviewed by the journal, your manuscript was reviewed by members of the editorial board and by several independent reviewers. In light of the reviews (below this email), we would like to invite the resubmission of a significantly-revised version that takes into account the reviewers' comments.

As you can see from the attached reviews, every reviewer recognized your study as important to the field and clearly appreciated the work you invested into writing and preparing this manuscript. Nevertheless, while all reviewers showed excitement over the novel BAC and CCMV model system you have created, some issues were raised that need to be addressed. There are two points in particular that were brought up during the review process that I would like to highlight. First, while all reviewers recognized the importance of a CCMV BAC, the question was raised as to how far this BAC actually reflects circulating WT CCMV isolates as multiple genes were lost during the cloning process. Furthermore, the BAC additionally acquired an unusual amount of sequence differences compared to the original viral sequence, either potentially indicating genome instability or sequencing errors. Reviewer 1 also raised additional concerns arising from the sequences differences between the generated BAC and the original parental virus, hence the authors should re-evaluate their identified mutations and examine potential effect these adaptations could have on the resulting BAC derived virus for viral genes identified as possibly affected by the reviewer. Secondly, both reviewer 1 and reviewer 3 noted that the experiments performed to characterize the CCMV BAC derived virus were actually performed on human cells, hence in cross-species infections. As explained in the reviews, this can be suboptimal as this could make the interpretation of the generated results more complicated as the virus is not adapted to this host species. Reviewer 3 suggested potentially useful approaches that authors could attempt to address this issue.

We cannot make any decision about publication until we have seen the revised manuscript and your response to the reviewers' comments. Your revised manuscript is also likely to be sent to reviewers for further evaluation.

Sincerely,

Daniel Malouli, Ph.D.

Guest Editor

PLOS Pathogens

Klaus Früh

Section Editor

PLOS Pathogens

Kasturi Haldar

Editor-in-Chief

PLOS Pathogens

orcid.org/0000-0001-5065-158X

Michael Malim

Editor-in-Chief

PLOS Pathogens

orcid.org/0000-0002-7699-2064

As you can see from the attached reviews, every reviewer recognized your study as important to the field and clearly appreciated the work you invested into writing and preparing this manuscript. Nevertheless, while all reviewers showed excitement over the novel BAC and CCMV model system you have created, some issues were raised that need to be addressed. There are two points in particular that were brought up during the review process that I would like to highlight. First, while all reviewers recognized the importance of a CCMV BAC, the question was raised as to how far this BAC actually reflects circulating WT CCMV isolates as multiple genes were lost during the cloning process. Furthermore, the BAC additionally acquired an unusual amount of sequence differences compared to the original viral sequence, either potentially indicating genome instability or sequencing errors. Reviewer 1 also raised additional concerns arising from the sequences differences between the generated BAC and the original parental virus, hence the authors should re-evaluate their identified mutations and examine potential effect these adaptations could have on the resulting BAC derived virus for viral genes identified as possibly affected by the reviewer. Secondly, both reviewer 1 and reviewer 3 noted that the experiments performed to characterize the CCMV BAC derived virus were actually performed on human cells, hence in cross-species infections. As explained in the reviews, this can be suboptimal as this could make the interpretation of the generated results more complicated as the virus is not adapted to this host species. Reviewer 3 suggested potentially useful approaches that authors could attempt to address this issue.

Reviewer's Responses to Questions

**Part I - Summary**

Reviewer #1: This study generates a BAC clone of a chimp CMV strain, then characterises it genomically, as well as by transcriptomics and proteomics. Given that chimp CMV is the closest relative to human CMV, studying the two can provide novel information, and this will be greatly facilitated by having a BAC clone.

Reviewer #2: In this lovely 'powerhouse' manuscript, the authors describe the BAC cloning and careful proteomic and transcriptome characterization of the chimpanzee cytomegalovirus (CCMV), which is the closest known animal virus relative to human cytomegalovirus (HCMV). The authors interpret their data cautiously, while also presenting a comparative analysis against data from the Weekes' laboratory. Their analyses adds enormous value to what would already be a highly appealing resource paper. HCMV is an important human pathogen that has also proven to be a source of myriad insights into molecular and cell biology (promoters, US11 / ERAD) and the immune system (ULBP's etc, NK cell activation and viral evasion thereof, et cetera). Until the work described in this study, there have not been any available reverse genetics system established to study CCMV, which is frankly shocking. Although Rhesus CMV (RhCMV) is an established primate model system, and the use of chimpanzees as model systems for biomedical research is no longer permitted, the comparative value of CCMV cannot be overstated-- especially since many RhCMV paralogs are functionally divergent from HCMV.. As the authors discuss, even certain CCMV gene products are not shared in HCMV and paralogs like UL148 exhibit only limited features in common. Not to get lost in the weeds here, having the CCMV genome cloned as a BAC now allows for careful examination of the acquisition of evolutionary novelty and divergent features, which will may in turn shed light on immune differences between primates.. as reflected by the viruses that infect them. I very rarely encounter such carefully prepared manuscripts.

My only very minor critique of the work is that on lines 224-243 they go over the temporal classes for different viral genes -- based on their expression. I never understood why Mike Weekes' otherwise excellent temporal viromics paper that established these classes for HCMV did not accurately encompass IE2-86 kd (UL122). This gene product is expressed both as an IE gene that rises and falls at very early times post infection (~4 hpi - 12 hpi) but then rises again as viral DNA synthesis commences, and keeps increasing up to a plateau that is maintained until the end of lytic infection. It's not just phenomenology because we know IE2 is critical for transactivation of E genes.. and IE2 is transiently expressed in abortive reactivation from latency.. the second phase of IE2 expression accompanies lytic progression. It may well not be appropriate for these authors to find a temporal category or class that fits this important gene.. but I thought I'd at least mention it here.

Congratulations to the authors. I am happy to sign this review.

Jeremy Kamil

Reviewer #3: CCMV is the closest known relative of HCMV with its genome highly collinear with that of HCMV. The sequencing data of CCMV were utilized to assess the coding potential of wild-type HCMV in early 2000 by Dr. Andrew Davison. CMVs are highly adapted to their host species and cross-species infections of CMVs are extremely unlikely to happen. Therefore, in vivo examination of all aspects of CMV biology relies on animal models and their respective CMVs. The most commonly used models for CMV are mice, rats, guinea pigs, and rhesus macaques. The close evolutionary relationship of nonhuman primates to humans is mirrored in the evolutionary relationship of the primate CMV genomes to HCMV. Among them, CCMV shares the highest level of similarity in genomic structure, coding capacity, and sequence homology. However, CCMV has never been considered as a model virus for HCMV due to the fact of chimpanzee models being phased out in most areas of research.

In this manuscript entitled “Engineering, Decoding and System-Level Characterization of Chimpanzee Cytomegalovirus”, the authors cloned the CCMV genome into BAC for the first time and subsequently repair the genome to near full-length. They also conducted the functional experiment demonstrating that, by repairing the frameshift mutation in UL128, productive CCMV infection could be re-established in endothelial and epithelial cells. In the second half of the manuscript, the authors re-evaluated the CCMV coding potential by systemically analyzing the transcriptome and proteome of infected cells and identified 14 novel ORFs, splice variants, and regulatory RNAs. Further, k-means clustering analyses of viral and host gene expression indicating that viral proteins cluster into five distinct temporal classes and host response to CCMV infection are in line with known hallmarks of HCMV infection.

Overall, this manuscript is well written and all experiments shown here were well executed. Although the transcriptome and proteome analyses part of this manuscript is very descriptive, I like the authors’ attempts to incorporate the datasets from other studies in the data analyses. Having said that, I strongly feel that conducting all experiments in human fibroblasts markedly reduced the biological significance of the data shown here.

Co-evolution of CMV with their host species over millions of years has led to some unique interactions between viral proteins and host intracellular/extracellular proteins (of various cell types). In the past, primate CMVs (chimpanzee, baboon, rhesus macaques) were isolated and adapted in cell cultures using human fibroblasts due to their readily availability. Chimpanzee fibroblasts are now possible to obtain and have been shown to have unique gene expressing profiles than their human counterparts. The authors should consider including some comparative analyses of CCMV infection in chimpanzee fibroblasts. Otherwise, one may suspect the discrepancies of temporal profiles (of viral gene expression) and their impacts on host cells (at both RNA and protein levels) between HCMV and CCMV are experimental artifacts resulting from CCMV replicating in cells not from its natural host.

**Part II – Major Issues: Key Experiments Required for Acceptance**

Reviewer #1: The inability to re-insert IRS1-TRS1 is disappointing, although a well recognised risk with inserting BAC cassettes into this region, based on previous experience with HCMV BACs. Does the genetic deletion here effect IRS, i.e. does their virus still form four genomic isoforms? Most importantly, do genes that may have transcriptional termini in this region (e.g. US2/3/6 etc) express properly? One of the reasons I ask is that US2 etc seem to express very early, then drop off almost instantly, which seems very odd for genes that play a major role in immune evasion throughout the virus lifecycle. Is this due to the lack of the IRS1-US1 region? Does the virus downmodulate MHC-I in the same way as the parental virus?

The power of the CCMV virus as a model system for HCMV, is when the virus is used in its natural cell type. Just because a virus grows in a cell line from another species doesn’t mean it modulates it in the same way. When used in human cell lines, there’s no way of knowing whether the presence or absence of any effects is because of species specificity. Does their virus infect chimp cell lines – I believe skin fibroblasts are available commercially?

According to the authors, their BAC clone has over 200 de novo mutations compared to the parental strain. This seems incredibly high. A number of studies have used NGS to show that HCMV BACs are extremely stable over time. Similarly in Line 180, their repaired BAC clone has a number of additional mutations that are not in the region that was modified. If the BAC clone is not stable, this is a major concern, and is not in line with other studies? Does it indicate a fundamental problem with genomic stability? Can they show that their BAC doesn't acquire random mutations when being propagated and modified?

Reviewer #2: None needed. N/A

Reviewer #3: Results Figure 3: Panels B and C. Please include CCMV (Heberling strain and BAC-derived) infection in chimpanzee fibroblasts.

Results Figure 6, 7, 8: If not possible to have comprehensive analyses in chimpanzee fibroblasts, the authors should conduct kinetic analyses of select CCMV genes in chimpanzee fibroblasts.

**Part III – Minor Issues: Editorial and Data Presentation Modifications**

Reviewer #1: Line 140. The authors call mutations as silent/non-coding etc based on the original CCMV annotation. Is there any risk that they have missed coding mutations in ORFs that were not annotated on the original sequence?

Can the applicants provide a schematic of the genome, with all known & novel ORFs/miRNA etc marked?

When restoring the US region, this is not just about making the BAC useful for immunological research. This is about making it a virus genome that actually expresses the phenotype of the real CCMV virus. Furthermore, other HCMV studies have shown that when the BAC cassette is not self-excised, it leads to genome instability when virus is reconstituted. These points need making too.

Could the applicants add some general characterisation of their proteomics profiling – e.g. how many viral/host genes were detected with >1 unique peptide. What proportion of host genes were modulated by the virus?

In the discussion of temporal classes, a slightly more thorough analysis of how CCMV and HCMV compare would be useful. For example how much overlap is there in other classes? I’m struck by how short the timeframe of expression of the TP1 & TP2 proteins is, and also by the fact that IE1 is apparently TP2 in CCMV? Is this expected? It’s not seen in HCMV?

In the discussion, they need to discuss the fact that they are using a chimp virus in human cells, and the limitations of doing this. They also need to make the point very clearly that they don’t know how well their BAC clone represents an authentic chimp CMV, since it has been passaged so extensively in human cells.

Fig 7. I’m not clear why the authors just compare the highest and lowest expressed genes. Can they not do a complete comparison of all proteins at any given timepoint?

I’m assuming that the WCL2 dataset in Table S5 has had proteins that are not in the CCMV dataset removed? If so, that needs stating.

In table S4/S5, I assume these are log2 values? If so, it needs stating. Similarly ‘relative’ needs to state what it’s relative to. E.g. for Table S4, I assume it’s relative to maximal expression. It also needs to state what ‘0’ means.

Discussion - self-excisable HCMV BACs have been constructed by many others prior to its application to TB40-BAC4, it would be good to cite those studies.

Reviewer #2: See main review. This is an excellent study.

Reviewer #3: Introduction: starting line 75, references update needed for the paragraph of RhCMV. A lot of progress has been made for the complete genomic sequence of RhCMV during the past 10 years, link below is the most recent reference: https://journals.plos.org/plospathogens/article?id=10.1371/journal.ppat.1008666

PLOS authors have the option to publish the peer review history of their article (what does this mean?). If published, this will include your full peer review and any attached files.

Reviewer #1: No

Reviewer #2: **Yes: **Jeremy P. Kamil

Reviewer #3: No
---

## [Decision Letter · Decision Letter 1]

9 Dec 2021

Dear Dr. Wiebusch,

We are pleased to inform you that your manuscript 'Engineering, decoding and systems-level characterization of chimpanzee cytomegalovirus' has been provisionally accepted for publication in PLOS Pathogens.

Best regards,

Daniel Malouli, Ph.D.

Guest Editor

PLOS Pathogens

Klaus Früh

Section Editor

PLOS Pathogens

Kasturi Haldar

Editor-in-Chief

PLOS Pathogens

orcid.org/0000-0001-5065-158X

Michael Malim

Editor-in-Chief

PLOS Pathogens

orcid.org/0000-0002-7699-2064

The reviewers conclusively stated that they are satisfied with the changes the authors have made to the manuscript. Minor changes should be addressed before publication, but other than that this manuscript seems to be ready for acceptance.

Reviewer Comments (if any, and for reference):

Reviewer's Responses to Questions

**Part I - Summary**

Reviewer #1: The reviewers have answered all my questions, thankyou! I just have 1 very minor text change, which I should have noticed in the previous version & could be dealt with at the proof stage if needed:

They mention the construction of BAC clones using alternative methods such as TAR cloning of synthetic fragments etc. A variant of this method has already been published for HCMV, and should be referenced: 10.1128/mSphereDirect.00331-17

**Part II – Major Issues: Key Experiments Required for Acceptance**

Reviewer #1: (No Response)

**Part III – Minor Issues: Editorial and Data Presentation Modifications**

Reviewer #1: (No Response)

PLOS authors have the option to publish the peer review history of their article (what does this mean?). If published, this will include your full peer review and any attached files.

Reviewer #1: No

---

## [Editor Report · Acceptance letter]

27 Dec 2021

Dear Dr. Wiebusch,

We are delighted to inform you that your manuscript, "Engineering, decoding and systems-level characterization of chimpanzee cytomegalovirus," has been formally accepted for publication in PLOS Pathogens.

Best regards,

Kasturi Haldar

Editor-in-Chief

PLOS Pathogens

orcid.org/0000-0001-5065-158X

Michael Malim

Editor-in-Chief

PLOS Pathogens

orcid.org/0000-0002-7699-2064